# A Glimpse into Photodetachment Spectra of Giant and Nested Fullerene Anions

**Valeriy K. Dolmatov** [1,*,†,‡] **and Steven T. Manson** [2,‡]

1    Department of Chemistry and Physics, University of North Alabama, Florence, AL 35632, USA
2    Department of Physics and Astronomy, Georgia State University, Atlanta, GA 30030, USA
*    Correspondence: vkdolmatov@una.edu
†    Retired. Current address: 1057 Dixie Ave., Florence, AL 35630, USA.
‡    These authors contributed equally to this work.

**Abstract:** We focus on the study of the photodetachment of bare, i.e., single-cage $(C_N)^-$ as well as nested (multi-cage) $(C_N@C_M@\dots)^-$ singly charged fullerene anions. We calculate the attached electron's wavefunctions, energies, oscillator strengths and photodetachment cross sections of the $C_{60}^-$, $C_{240}^-$, $C_{540}^-$, $(C_{60}@C_{240})^-$, $(C_{60}@C_{540})^-$, $(C_{240}@C_{540})^-$ and $(C_{60}@C_{240}@C_{540})^-$ fullerene anions, where the attached electron is captured into the ground *s*-state by the resultant external field provided by all fullerene cages in the anion. The goal is to gain insight into the changes in behavior of photodetachment of this valence electron as a function of the different geometries and potentials of the various underlying fullerenes or nested fullerenes (fullerene onions) both due to their increasing size and due to "stuffing" of a larger bare fullerene with smaller fullerenes. To meet this goal, we opt for a simple semi-empirical approximation to this problem: we approximate each individual fullerene cage by a rigid potential sphere of a certain inner radius, thickness and potential depth, as in numerous other model studies performed to date. The results reveal a number of rather significant differences in the wavefunctions, oscillator strengths and photodetachment cross sections among these fullerene anions, some of which are completely counter-intuitive. The results obtained can serve as a "zeroth-order-touchstone" for future studies of single-cage and nested fullerene anions by more rigorous theories and/or experiments to build upon this work to assess the importance of interactions omitted in the present study.

**Keywords:** photodetachment; carbon fullerenes; carbon fullerene anions; carbon fullerene onions





## 1. Introduction

The photoionization/photodetachment of various neutral ($q = 0$) and charged ($q \neq 0$) fullerenes, $C_N^{\pm q}$, and their endohedral counterparts, $A@C_N^{\pm q}$ (where $A$ is the atom encapsulated inside $C_N^{\pm q}$ cage), has been the subject of experimental (see, e.g., [1–7] and references therein) as well as intense systematic theoretical studies for many years now (see, e.g., a recent review paper [8] with an abundance of references therein). In particular, Professor M.Y. Amusia, to the legacy of whom this Special Issue of *Atoms* is devoted to, has contributed vastly to the study of the interaction of particles and light with fullerenes and endo-fullerenes, see, e.g., [9–18], to name a few.

Although the research on the interaction of fullerenes and endo-fullerenes with light has also touched upon the subject of fullerene anions (see, e.g., [1–4,8,9,19–24] and references therein), yet, to the best of the authors' knowledge, the subject of photodetachment of giant fullerenes anions [$(C_N)^-$ with $N \gg 60$] as well as of nested fullerene anions, $(C_n@C_{m>n}@\dots)^-$, referred to as fullerene onion-anions in the present paper, has not been studied. Given the current strong interest in studying various elementary processes of basic importance involving fullerene formations, it is appealing to fill in this gap in the present state of knowledge. The present paper remedies the situation by presenting a first

insight into the phenomenon of photodetachment of both giant fullerene and fullerene onion-anions.

In general, elementary processes involving fullerene formations present a formidable multifaceted problem for theorists, thereby requiring the investment of considerable efforts to comprehensibly address all facets of the problem as well as the interaction(s) between them. Therefore, before investing such efforts in a comprehensive study, a kind of roadmap is needed as a guide to the subsequent comprehensive study of this multifaceted problem. Thus, the main narrow goal of the present study is to gain insight, using the simplest reasonable approximation, for modifications of the photodetachment cross sections of giant and nested fullerene anions owing to changes in their geometry induced by stuffing of a larger bare fullerene with smaller and smaller fullerenes: $(C_{240})^-$, $(C_{540})^-$, $(C_{60}@C_{240})^-$, $(C_{60}@C_{540})^-$, $(C_{240}@C_{540})^-$ and $(C_{60}@C_{240}@C_{540})^-$. To meet this goal, we approximate each individual fullerene cage by a rigid potential sphere of a certain inner radius $r_{in}$, thickness $\Delta$ and potential depth $U_0$, as in many earlier model studies of fullerene-involved processes cited above. Within the framework of this approximation, we detail how the photodetachment of the fullerene onion-anions differs crucially from the photodetachment of the largest bare (i.e., single-cage) fullerene anion owing to the differences in geometries between the fullerene formations.

To label the discrete states occupied by the attached electron in a fullerene anion, we adopt, just as a matter of labeling, the traditional notation used for atoms, i.e., the $n\ell$-notation, where $\ell$ is the orbital quantum number and $n \geq \ell + 1$. Thus, in our notations, the first $s$-state of the attached electron is $1s$, the next $s$-state is a $2s$ state, the first $p$-state is a $2p$ state, the next $p$-state is a $3p$ state, and so on.

Finally, atomic units (a.u.) ($|e| = \hbar = m = 1$, where $e$ and $m$ are the electron's charge and mass, respectively, and $\hbar$ is a reduced Planck's constant) are used throughout the paper unless stated otherwise.

## 2. Review of Theory

We model an individual $C_N$ cage ($N$ being the number of carbon atoms in the cage) by a $U_{C_N}(r)$ spherical annular potential of the inner radius, $r_{in}$, finite thickness, $\Delta$, and depth, $U_0$:

$$U_{C_N}(r) = \begin{cases} -U_0, & r_{in} \leq r \leq r_{in} + \Delta \\ 0, & \text{otherwise.} \end{cases} \tag{1}$$

Such modeling of a $C_N$ cage was suggested in the early work by Puska and Nieminen [25] and, since then, has found an extensive use in numerous studies to date; the reader is referred to [5,13,16,18,21,22,25–27] and to the review paper [8] for many more references on the subject, as well as, e.g., to references [6,8–10,18,21,22,29,31,46] from [21] (and references therein).

We emphasize that, with regard to $C_{60}$, such model has been proven [5,18,26] (and references therein) to produce results in a reasonable agreement with the experimental photoionization spectrum of endohedral Xe@$C_{60}^+$ [5] and a qualitative and even semi-quantitative agreement with experimental differential elastic electron scattering off $C_{60}$ [6]. Such modeling was also shown [27] to result in a semi-quantitative agreement with some of the most prominent features of the $e^- - C_{60}$ total elastic electron scattering cross section predicted by a far more sophisticated ab initio molecular-Hartree–Fock approximation [27]. This lays out a supporting background for a reasonable suitability of such modelling of $C_{60}$ for the application to photodetachment of a $C_{60}$ fullerene anion as well. Furthermore, our model replaces the earlier fullerene-anion-photodetachment approximations [9,19], which utilized the idea of an infinitesimally thin fullerene wall, by a more realistic finite-width-wall approximation, which is certainly an improvement to the cited approximations.

A fullerene onion, then, is modeled by a potential which is a linear combination of the corresponding $U_{C_N}(r)$ potentials, as in [28]:

$$U_{C_N}@C_M@\ldots = U_{C_{60}} + U_{C_{240}} + \ldots. \tag{2}$$

The parameters $r_{in}$, $U_0$ and $\Delta$ of the individual $C_{60}$, $C_{240}$ and $C_{540}$ fullerene cages in fullerene onion-anions are assumed to be the same as for the corresponding isolated bare (single-cage) fullerenes. In the present paper, we take the values for $r_{in}$, $\Delta$ and $U_0$ for $C_{60}/C_{240}/C_{540}$ from [28]: $r_{in} = 5.8/12.6/18.8$, $\Delta = 1.9/1.9/1.9$ and $U_0 = 8.22/10/12$ eV, respectively.

A fullerene anion, $C_N^-$, or a fullerene onion-anion, $(C_N@C_M@\ldots)^-$, then, is formed by binding of an external electron into a $s$-state or a $p$-state in the field of corresponding $U_{C_N}(r)$ or $U_{C_N@C_M@\ldots}$ potential, respectively. Thus, the bound, $P_{n\ell}$, and continuum, $P_{\epsilon\ell}$, radial wavefunctions for the attached electron in a corresponding fullerene anion satisfy the radial Schrödinger equation:

$$-\frac{1}{2}\frac{d^2 P_{n/\epsilon\ell}}{dr^2} + \left[ U_C + \frac{\ell(\ell+1)}{2r^2} \right] P_{n/\epsilon\ell}(r) = E_{n/\epsilon\ell}P_{n/\epsilon\ell}(r). \tag{3}$$

Here, $n$ and $\ell$ are is the principal and orbital quantum numbers, respectively, $\epsilon$ is the photoelectron energy and $U_C$ is the fullerene potential determined by Equations (1) or (2), respectively.

This equation is solved with the following boundary conditions for the discrete and continuum states:

$$P_{n\ell}(r)|_{r\to 0,\infty} = 0, \text{ whereas } P_{\epsilon\ell}(r)|_{r\gg 1} \to \sqrt{\frac{2}{k\pi}}\sin\left(kr - \frac{\pi\ell}{2} + \delta_\ell(\epsilon)\right). \tag{4}$$

Here, $\delta_\ell(\epsilon)$ is the phase of the continuum state wavefunction and $k$ is the photoelectron momentum.

Note that such model of fullerene anion photodetachment is similar in spirit to the one suggested earlier [9,19], albeit there is a Dirac-bubble potential, rather than the spherical annular potential, was used to approximate the $C_{60}$ cage.

The photodetachment cross sections, $\sigma_{n\ell\to\epsilon,\ell\pm 1}$, as well as the oscillator strengths of the discrete, $f_{n\ell\to n',\ell\pm 1}$, and continuum, $f_{n\ell\to\epsilon,\ell\pm 1}$, spectra of fullerene anions, were calculated using well-known formulas, see, e.g., [29]:

$$\sigma_{n\ell\to\epsilon,\ell\pm 1} = \frac{4}{3}\pi^2\alpha\frac{N_{n\ell}}{2\ell+1}\omega d_{\ell\pm 1}^2, \tag{5}$$

$$f_{n\ell\to n',\ell\pm 1} = \frac{N_{n\ell}}{3(2\ell+1)}\omega d_{\ell\pm 1}^2, \tag{6}$$

$$f_{n\ell\to\epsilon,\ell\pm 1} = \frac{1}{2\pi^2\alpha}\int_0^\infty \sigma_{n\ell\to\epsilon,\ell\pm 1}d\omega. \tag{7}$$

Here, $\alpha$ is the fine-structure constant, $\omega$ is the photon energy, $N_{n\ell}$ is the number of electron in the $n\ell$ state (a single electron in our case), and $d_{\ell\pm 1}$ is the reduced radial matrix element for the transition from the $n\ell$ state to a $n'(\epsilon), \ell\pm 1$ final state.

## 3. Results and Discussion

### 3.1. Single-Cage Fullerene Anions

As the first step, we scrutinize the $1s$ ground-states and $2p$ excited-states in the bare fullerene anions: $C_{60}^-$, $C_{240}^-$ and $C_{540}^-$. We note that our calculations revealed no $np$ excited-states with $n > 2$ in any of these anions. The corresponding ground-state $P_{1s}(r)$ and

excited-state $P_{2p}(r)$ radial functions and the corresponding $E_{1s}$ and $E_{2p}$ energies of the attached electron in the $C_{60}^-$, $C_{240}^-$ and $C_{540}^-$ anions are presented in Figure 1.

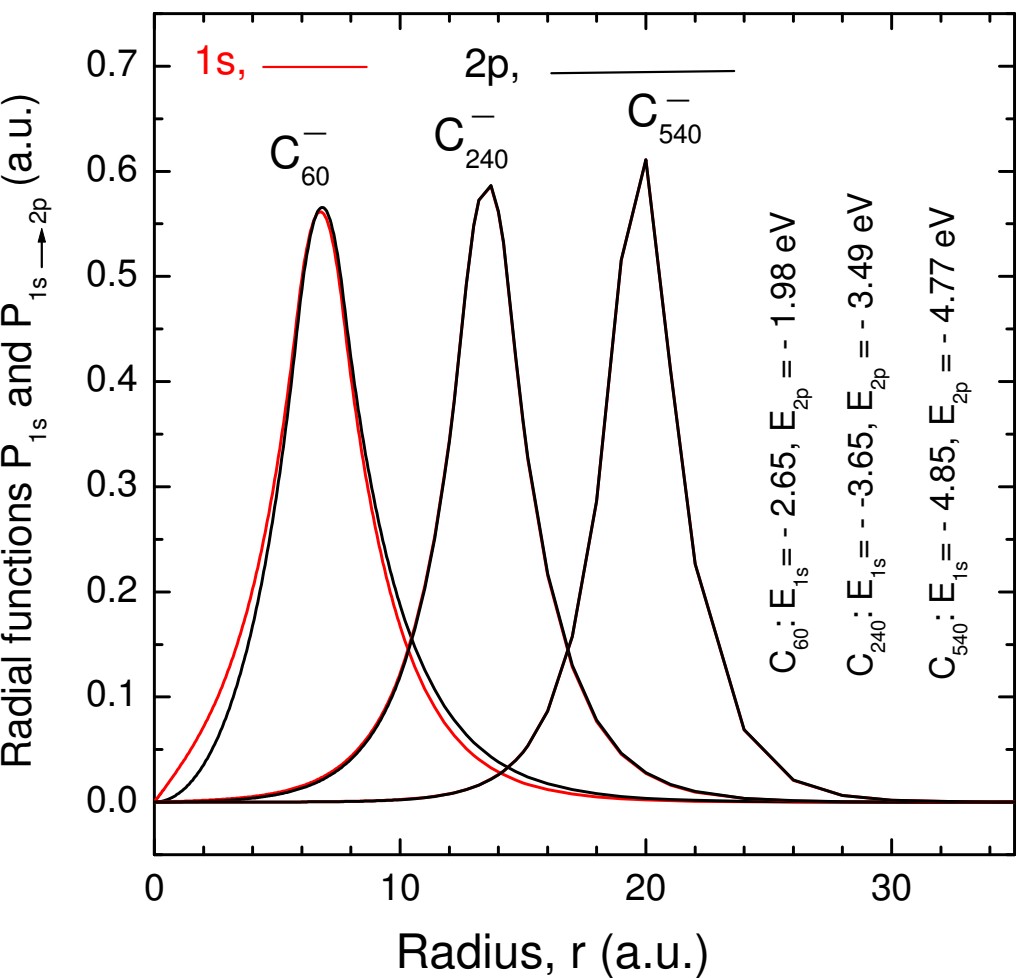

**Figure 1.** Calculated ground-state $P_{1s}(r)$ and excited- state $P_{2p}(r)$ radial functions and corresponding $E_{1s}$ and $E_{2p}$ binding energies of the attached electron in $C_{60}^-$, $C_{240}^-$, and $C_{240}^-$ anions, as designated.

One can see that the $P_{1s}$ and $P_{2p}$ functions reach their maxima within the wall of a corresponding fullerene, i.e., within $5.8 < r < 7.7$ in $C_{60}$, $12.6 < r < 14.5$ in $C_{240}$ and $18.85 < r < 20.75$ in $C_{540}$. This, actually, has been expected, for an obvious reason. A strikingly unexpected result (at first glance), though, is that $P_{1s} \approx P_{2p}$ to a high degree of approximation, particularly in $C_{240}^-$ and $C_{540}^-$. This seems strange, because the Schrödinger equations for a *s*-state and a *p*-state differ by the presence of a centrifugal potential $U_{\mathrm{cfg}} = \frac{\ell(\ell+1)}{2r^2}$ for a *p*-state. Correspondingly, the $P_{1s}$ function should have differed from the $P_{2p}$ function. To understand why the situation is opposite to the expected one, we depict, for the case of $C_{60}$, the cage model potential $U_{C_{60}}(r)$, the centrifugal potential $U_{\mathrm{cfg}} = \frac{\ell(\ell+1)}{2r^2} = \frac{1}{r^2}$ for a *p*-electron, and the 2*p* probability density distribution, $\rho_{2p} = P_{2p}^2$, in Figure 2.

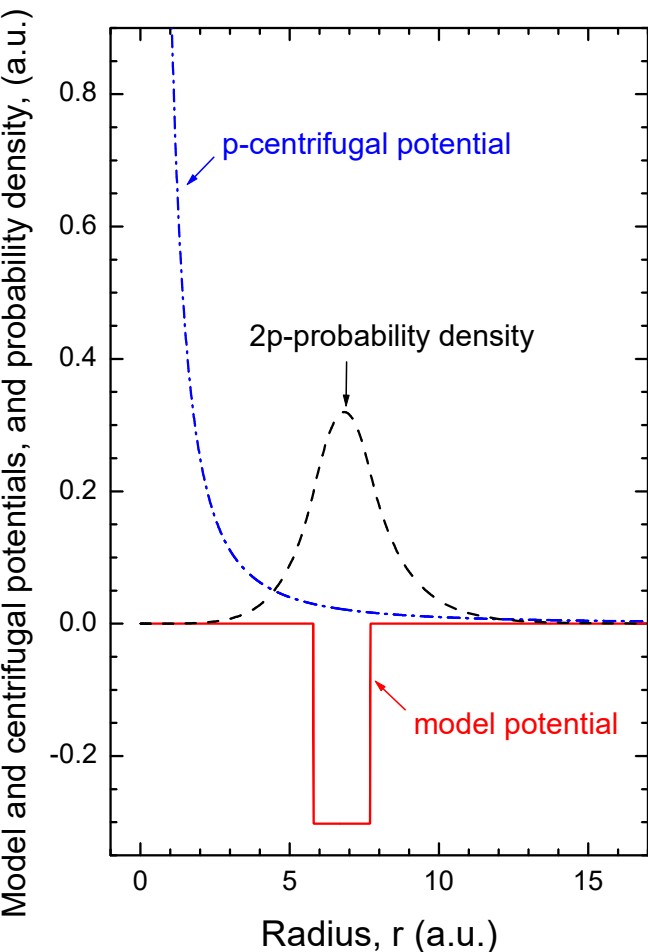

**Figure 2.** Calculated cage model potential $U_{C_{60}}(r)$, the centrifugal potential $U_{cfg} = \frac{\ell(\ell+1)}{2r^2} = \frac{1}{r^2}$ for a *p*-electron, and the 2*p* probability density distribution, $\rho_{2p} = P_{2p}^2$, as designated.

One can see that, inside the hollow interior of $C_{60}$ ($U_{C_{60}} = 0$), the $\rho_{2p}$ probability density is practically a zero up to about *r* = 3. Therefore, the presence of the centrifugal potential, however large it is, does not matter in this spatial region. $\rho_{2p}$ starts differ from $\rho_{2p} \approx 0$ between approximately $3 < r < 5.8$. There, however, $U_{cfg}$ is already small and, additionally, only less than 20% of electronic charge is concentrated in this spatial region. Hence, again, a role of the small $U_{cfg} \neq 0$ is largely obliterated in there. Inside the $C_{60}$ wall itself $U_{cfg}$, on average, is less than 3% of $U_{C_{60}} = 0.302$, whereas outside of the $C_{60}$ wall, $U_{cfg} \ll U_{C_{60}}$ in addition to a rapidly damping probability density distribution. Thus, it now becomes clear that the presence of the centrifugal potential for a *p*-electron cannot make the solution of the Schrödinger equation to differ any notably from its solution for a *s*-electron. This discussion explains why $P_{1s}$ differs from $P_{2p}$ only insignificantly, in $C_{60}^-$. Additionally, we believe that the reader can easily extend this discussion of the behavior of $P_{1s}$ and $P_{2p}$ in $C_{60}^-$ to giant fullerene anions to understand why $P_{1s}$ and $P_{2p}$ become practically identical in each of the $C_{240}^-$ and $C_{540}^-$ anions.

Although the energies are generally more sensitive quantities to parameters in the Schrödinger equation than the wavefunctions, the difference between $E_{1s}$ and $E_{2p}$ binding energies is more noticeable than between the wavefunctions, although still small: it is about 25% of $E_{1s}$ for $C_{60}^-$, 4% for $C_{240}^-$, and 1.6% for $C_{540}^-$. Note that the difference between $E_{1s}$ and $E_{2p}$ is decreasing with increasing size of the fullerene anion. The largest energy difference 25% is in $C_{60}^-$, as is the largest difference between $P_{1s}$ and $P_{2p}$ (see Figure 1). This is because the 2*p*-centrifugal potential energy in $C_{60}^-$ is larger than in other fullerene anions, owing to a significantly smaller size of the $C_{60}$ cage as compared to the other two.

Because $P_{1s} \approx P_{2p}$, the corresponding $f_{1s \to 2p}$ oscillator strengths in the $C_{60}^-$, $C_{240}^-$ and $C_{540}^-$ anions must be large. Our calculations show that $f_{1s \to 2p} \approx 0.807$, 0.962, and 0.679 in $C_{60}^-$, $C_{240}^-$ and $C_{540}^-$, respectively (see Table 1 for more details).

**Table 1.** Calculated $E_{1s}$ ground-state energies, $\omega_{2p}$ and $\omega_{3p}$ energies of the $1s \to 2p\&3p$ transitions (all in eV), discrete $f_{1s \to 2p\&3p}$, and continuum, $f_{1s \to \epsilon p}$, oscillator strengths in the single-cage and multi-cage fullerene anions. Note, our calculations showed no existence of the $np$ excited states with $n > 2$ in the single-cage fullerene anions.

| Anions | $E_{1s}$ | $\omega_{2p}$ | $\omega_{3p}$ | $f_{1s \to 2p}$ | $f_{1s \to 3p}$ | $f_{1s \to (2p+3p)}$ | $f_{1s \to \epsilon p}$ |
|---|---|---|---|---|---|---|---|
| $C_{60}^-$ | $-2.654$ | 0.6743 | - | 0.807 | - | 0.807 | 0.194 |
| $C_{240}^-$ | $-3.646$ | 0.155 | - | 0.692 | - | 0.692 | 0.307 |
| $C_{540}^-$ | $-4.855$ | 0.0831 | - | 0.6795 | - | 0.6795 | 0.326 |
| $(C_{60}@C_{240})^-$ | $-3.691$ | 0.1782 | 1.8314 | 0.7345 | 0.025 | 0.7595 | 0.239 |
| $(C_{60}@C_{540})^-$ | $-4.855$ | 0.0831 | 2.8902 | 0.6795 | 0.000 | 0.6796 | 0.324 |
| $(C_{240}@C_{540})^-$ | $-4.903$ | 0.092 | 1.514 | 0.696 | 0.052 | 0.748 | 0.260 |
| $(C_{60}@C_{240}@C_{540})^-$ | $-4.903$ | 0.093 | 1.483 | 0.699 | 0.052 | 0.751 | 0.256 |

We note that the oscillator strength $f_{1s \to 2p}$ is decreasing with increasing size of the fullerene cage. At first glance this is strange, because the approximate equality $P_{1s} \approx P_{2p}$ is getting only stronger with increasing size of the fullerene cage, as discussed above. Thus, the overlap between $P_{1s}$ and $P_{2p}$ is increasing and so should have been $f_{1s \to 2p}$ as well, with increasing size of the anion. However, the $\omega_{1s \to 2p} \equiv \omega_{2p}$ excitation energy (see Table 1), is decreasing with the increasing size of the fullerene cage. This counterbalances the increase in the overlap between $P_{1s}$ and $P_{2p}$, thereby resulting in a smaller $f_{1s \to 2p}$ (which is proportional to $\omega_{np}$) in a bigger fullerene anion. This decrease in the $f_{1s \to 2p}$ oscillator strength with increase in the fullerene size leads to an important conclusion. Namely, we conclude there is an increasing transfer of oscillator strength of a fullerene anion from a discrete spectrum to continuum with increasing size of the fullerene cage, as clearly follows from the oscillator strength sum rule: $f_{1s \to \epsilon p} = 1 - f_{1s \to 2p}$. Calculated $f_{1s \to \epsilon p}$'s are presented in Table 1 as well. At this point it is important to emphasize that the continuum oscillator strengths, presented in Table 1, were calculated using Equation (7) rather than as $1 - f_{1s \to 2p}$ from the sum rule. The fact that the independently calculated $f_{1s \to \epsilon p}$ and the $f_{1s \to \epsilon p} = 1 - f_{1s \to 2p}$ are equal to a high degree of approximation speaks about the adequacy of the calculated photodetachment cross sections themselves, discussed later in the paper.

*3.2. Fullerene Onion-Anions*

We now move to the discussion of the wavefunctions of the valence electron in the fullerene onion-anions $(C_{60}@C_{240})^-$, $(C_{60}@C_{540})^-$, $(C_{240}@C_{540})^-$ and $(C_{60}@C_{240}@C_{540})^-$. We note first that the potentials of these fullerene onions are, obviously, either double-well or triple-well potentials. Correspondingly, one can expect a greater number of bound states available to the attached electron in these fullerene onion-anions. In our case, the calculations predicted the existence of only two discrete $p$-states—the $2p$ and $3p$ excited states—, in contrast to only the $2p$ excited-state in the bare fullerene anions. The corresponding $P_{1s}$, $P_{2p}$ and $P_{3p}$ functions are plotted in Figure 3 where a number of new features are exhibited.

The most striking discovery relates to the behavior of the $P_{3p}$ excited-state wave functions. Their highly peculiar behavior is completely different from the behavior of the $P_{1s}$ and $P_{2p}$ functions in any of these fullerene onion-anions. Indeed, we find that a significant part of the $P_{3p}$ function and, thus, the electron density of the attached electron, is packed inside the wall of the inner cage directly adjacent to a larger fullerene cage in each of these double- and triple-cage fullerene onion-anions. Although this is evident from Figure 3, this is also supported by looking at the mean radii, $\bar{r}_{3p}$, of the $3p$ orbitals in these fullerene onion-anions as well. The calculated $\bar{r}_{3p}$'s are: $\bar{r}_{3p} \approx 7$ in both $(C_{60}@C_{540})^-$

and $(C_{60}@C_{240})^-$ (thus, the $3p$ orbital falls into the $C_{60}$ potential well), whereas $\bar{r}_{3p} \approx 14$ in both $(C_{240}@C_{540})^-$ and $(C_{60}@C_{240}@C_{540})^-$ (thus, $\bar{r}_{3p}$ falls into the $C_{240}$ potential well). This is in a sharp contrast to the $P_{1s}$ and $P_{2p}$ functions that are mainly packed in the potential well of the largest fullerene cage ($C_{540}$, in our case) in corresponding fullerene onion-anions, respectively, as is evident from Figure 3 (also, $\bar{r}_{1s} \approx \bar{r}_{2p} \approx 20$ in all fullerene onion-anions under discussion). Especially surprising is the behavior of the $P_{3p}$ function in $(C_{60}@C_{540})^-$, where its probability density is almost entirely located inside $C_{60}$, despite the size of $C_{60}$ being significantly smaller than $C_{540}$, so that the $C_{60}$ potential well should not have affected the attached valence electron at all, as in the case of the $P_{1s}$ and $P_{2p}$ functions (see Figure 3c,d). In any case, the behavior of $P_{3p}$ in these fullerene onion-anions is extraordinary, a complete break with conventional wisdom.

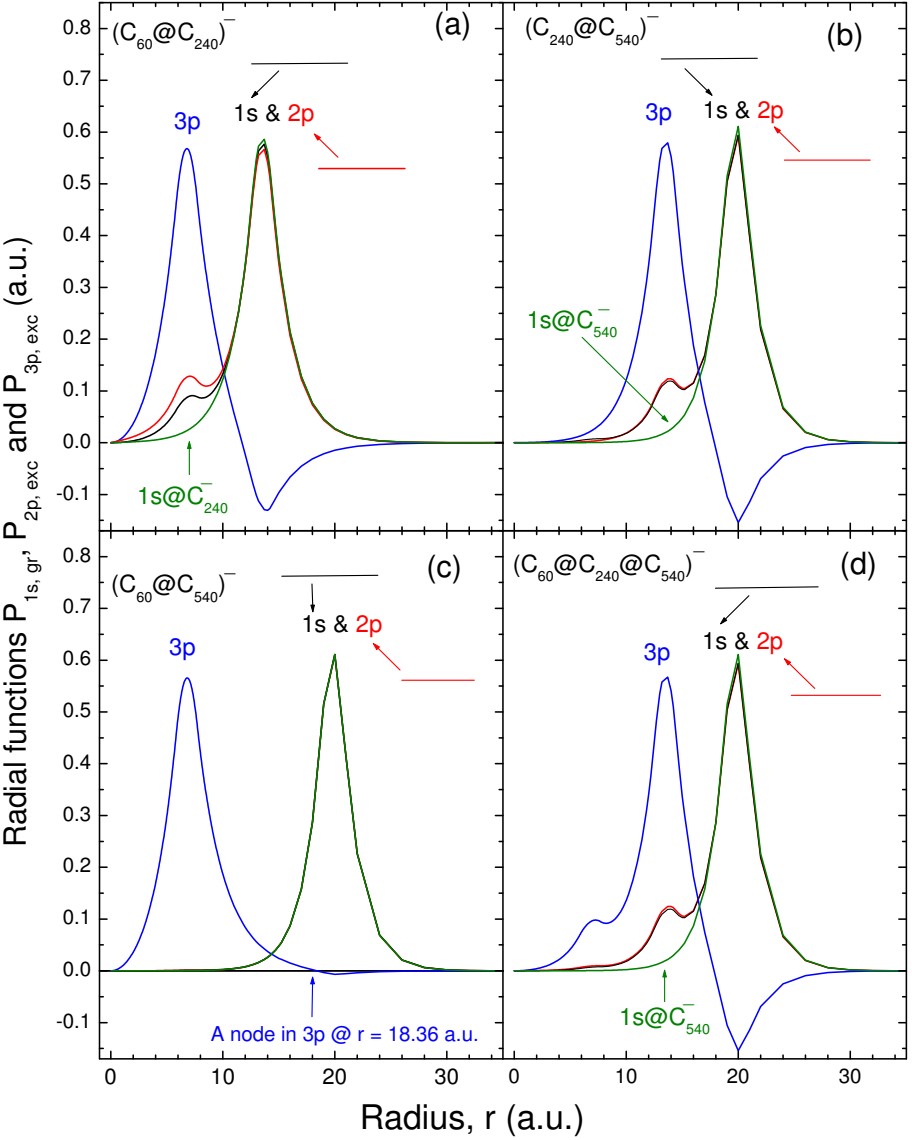

**Figure 3.** Calculated radial ground-state $P_{1s}(r)$ and excited-state $P_{2p}$ and $P_{3p}$ (due to the $1s \to np$ transitions, $n = 2, 3$) of the attached electron in fullerene onion-anions: (**a**) $(C_{60}@C_{240})^-$, (**b**) $(C_{240}@C_{540})^-$, (**c**) $(C_{60}@C_{540})^-$, (**d**) $(C_{60}@C_{240}@C_{540})^-$: solid, $P_{2p}$; dashed, $P_{1s}$; dash–dot, $P_{3p}$. Also plotted are the $P_{1s}$ (dash–dot–dot) functions in bare $C_{240}^-$ and $C_{540}^-$, as designated, for comparison purposes. Note, to avoid any confusion, that the graphs for the $P_{1s}$ and $P_{2p}$ functions in the fullerene onion-anions tightly overlap with each other and are practically indistinguishable from each other with some exception in the case of $(C_{60}@C_{240})^-$.

We interpret the predicted behavior of the $P_{3p}$ excited-state functions in the fullerene onion-anions as being due to both the multi-well nature of the fullerene onion-anion potentials and the fact that, in contrast to the nodeless $P_{1s}$ and $P_{2p}$ functions, the $P_{3p}$ function has one node. That is, the $P_{3p}$ function is distinctly split into an inner part (before the node) and an outer part (beyond the node). It appears that the inner part of the $P_{3p}$ function falls into the potential well associated with a fullerene cage adjacent to the outermost cage in the fullerene onion-anion. Thus, the attached electron partially resides in the inner well.

We note, though, that the behavior of the $P_{3p}$ function in the fullerene onion-anions is somewhat reminiscent of the behavior of the excited $P_{3d}$ and $P_{4d}$ functions, excited from the $3p$ subshell, in endohedral calcium, Ca@$C_{60}$ [30]. There, a significant transfer of the $4d$, but not $3d$, electron density into the inner space of $C_{60}$ was demonstrated. That resulted in a significant increase in the amplitude of the $P_{4d}$ orbital in the inner space of $C_{60}$. Consequently, the mean radius of the $4d$ orbital was reduced from $\bar{r}_{4d} \approx 14$ in free Ca to only $\bar{r}_{4d} \approx 4.3 < r_c = 5.8$ in Ca@$C_{60}$ [30]. That situation, in turn, was commented on to be somewhat reminiscent of the behavior of the excited $4f$ and $5f$ orbitals in Ba$^+$ [31,32] that was shown to be due to the double-well nature of the potential of Ba$^+$ that caused partial orbital collapse of $5f$ into the inner well, thereby causing $5f$, rather than $4f$, to have the greater amplitude near $r = 0$.

*3.3. Oscillator Strengths and Photodetachment Cross Sections*

Calculated oscillator strengths, $f_{1s \to np}$, of the $C_{60}^-$, $C_{240}^-$, and $C_{540}^-$ bare fullerene anions as well as the $(C_{60}@C_{240})^-$, $(C_{60}@C_{540})^-$, $(C_{240}@C_{540})^-$, and $(C_{60}@C_{240}@C_{540})^-$ fullerene onion-anions are/were listed in Table 1 which contains a wealth of information. Since a principal goal of this work is to explore the spectral distribution of oscillator strength, we focus on a comparison among the total oscillator strengths of the discrete spectra, i.e., $f_{1s \to 2p} + f_{1s \to 3p} \equiv f_{(2p+3p)}$, for the single and nested fullerene cages Thus, as we transition from $C_{240}^- \to (C_{60}@C_{240})^-$, the $f_{(2p+3p)}$ oscillator strength is increased. The same change in $f_{(2p+3p)}$ is characteristic along all other transition paths as well: $C_{540}^- \to (C_{240}@C_{540})^-$, $C_{540}^- \to (C_{60}@C_{240}@C_{540})^-$, and, in principle, $C_{540}^- \to (C_{60}@C_{540})^-$, too. Hence, we have unraveled a general tendency: stuffing of a bigger fullerene cage with a smaller fullerene cage, as well as progressively stuffing the biggest fullerene cage with several smaller fullerene cages, results in the transfer of a part of the oscillator strength from a continuum spectrum into a discrete spectrum.

Now, how does the discovered tendency affect the $1s$-photodetachment cross section, $\sigma_{1s}$? Obviously, the total area under the graph for $\sigma_{1s}$ should be decreasing along the discussed fullerene transition paths. This may result in the disappearance of some of the resonance structures in $\sigma_{1s}$, or making them narrower, or decreasing their heights, or lowering the values of other parts of $\sigma_{1s}$, or all of the above cumulatively. It is, therefore, extremely interesting to study the modifications in $\sigma_{1s}$'s on a comparative one-to-one basis for different fullerene anions.

Calculated $\sigma_{1s}$'s for $C_{240}^-$ versus $(C_{60}@C_{240})^-$, as well as $C_{540}^-$ versus $(C_{60}@C_{540})^-$, $(C_{240}@C_{540})^-$, and $(C_{60}@C_{240}@C_{540})^-$ are depicted in Figure 4 as functions of the photoelectron momentum $\kappa$, in order to eliminate the impact of differences in $1s$ ionization potentials between the fullerene anions on details of $\sigma_{1s}$'s, for the adequacy of the comparison between these anions.

We first note that the calculated cross sections exhibit the oscillatory behavior versus the photoelectron momentum, $k$. Such resonances have been well understood for both photoionization and photodetachment of, as well as electron scattering by, fullerene and endo-fullerene complexes in a large body of research; we refer the reader to the above references, to the review paper [8] for many more references on the subject, as well as, e.g., to [4–8,10–24,29,31,38,39,46] from [21] (and references therein). Following [33], these resonances are commonly referred to as the *confinement resonances*.

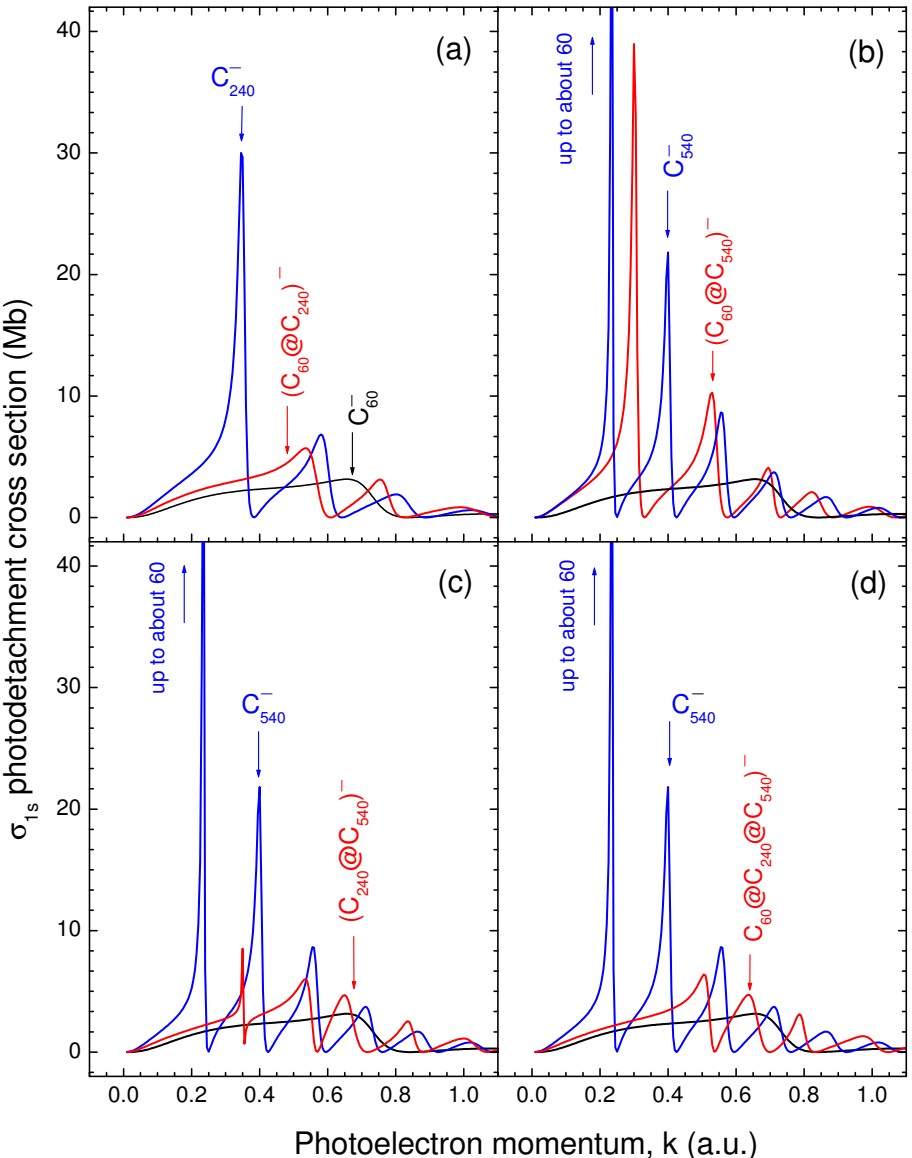

**Figure 4.** Calculated $\sigma_{1s}$ photodetachment cross sections of bare fullerene anions and nested fullerene onion-anions, as designated in the figure. Note, on all parts of the figure, the $\sigma_{1s}$ of $C_{60}^-$ is represented by a dashed-line.

Secondly, note that the prediction mentioned above on the modification of the photodetachment cross section along the path from the bare fullerene anions to the double- and triple-cage fullerene onion-anions is seen to be correct. Indeed, we see the disappearance of one or even two confinement resonance structures (near the lower energy end of the spectrum), and the significant decrease in their amplitudes (except for the case of the $(C_{60}@C_{540})^-$ onion-anion, which is an extraordinary case anyway, as was discussed above). Furthermore, it is interesting that the resonances in $\sigma_{1s}$ of $(C_{60}@C_{540})^-$ are seen to be shifted towards higher $k$'s, compared to $\sigma_{1s}$ of the bare $(C_{540})^-$ anion, whereas in all other nested fullerene onion-anions they shift toward lower $k$'s, compared to corresponding bare counterparts.

Thirdly, it is quite interesting that $\sigma_{1s}$'s of all fullerene onion-anions, whether double-cage or triple-cage anions, do not differ much in magnitude from $\sigma_{1s}$ of the smallest $C_{60}^-$ anion in this sequence of fullerene anions (except for $\sigma_{1s}$ of extraordinary $(C_{60}@C_{540})^-$ at a lower end of the spectrum). To emphasize this, we added $\sigma_{1s}$ of $C_{60}^-$ to all plots in Figure 4 to facilitate this comparison. One can see that $\sigma_{1s}$'s, associated with the nested

fullerene onion-anions, oscillate around $\sigma_{1s}$ of $C_{60}^-$ with average amplitudes that are not much different from $\sigma_{1s}$ of $C_{60}^-$.

Lastly, we note that, to check the connection of calculated $\sigma_{1s}$'s to calculated oscillator strengths, we calculated the oscillator strength of the continuum spectrum, $f_{1s \to \epsilon p}$, by appropriately integrating $\sigma_{1s}$'s in accordance with Equation (7). These calculated $f_{1s \to \epsilon p}$'s are presented in Table 1, and the fact that they have the same values as those obtained from the oscillator sum rule ($f_{1s \to \epsilon p} = 1 - f_{1s \to (2p+3p)}$) speaks to the accuracy of the calculated $\sigma_{1s}$'s.

## 4. Conclusions

In the present paper, we have provided a glimpse into the structure and photode-tachment cross sections of bare fullerene anions and nested fullerene onion-anions and uncovered the existence of a number of unusual features. The results were obtained on a zeroth-order basis, so to speak. However, the zeroth-order basis is a valuable and neces-sarily part of any study of any multielectron atomic and molecular systems and processes. This is because, firstly, it provides a kind of a roadmap where more sophisticated theoretical studies of these systems should be conducted and, secondly, the comparison between the results obtained with a more accurate calculation with those obtained in this zeroth-order study is the only way to understand the importance and strength of the physical interac-tions that are not accounted in the framework of the zeroth-order approximation. We hope that the results of the present study will serve as an impetus to more complete theoretical studies of the structure and photodetachment spectra of fullerene (onion–)anions, now that we know that they might be quite unusual.

**Author Contributions:** Conceptualization, V.K.D. and S.T.M.; methodology, V.K.D. and S.T.M.; formal analysis, V.K.D. and S.T.M.; investigation, V.K.D. and S.T.M.; writ-ing—original draft preparation, V.K.D. and S.T.M.; writing—review and editing, V.K.D. and S.T.M.; All authors have read and agreed to the pub-lished version of the manuscript.

**Funding:** The work of STM was supported by the US Department of Energy, Office of Science, Basic Energy Sciences under Award Number DE-FG02-03ER15428.

**Institutional Review Board Statement:** Not applicable.

**Informed Consent Statement:** Not applicable.

**Conflicts of Interest:** The author declares no conflict of interest.

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
