# Peer review of "A Glimpse into Photodetachment Spectra of Giant and Nested Fullerene Anions"

_atoms, doi:10.3390/atoms10040099_

Round 1

Reviewer 1 Report

In this work the authors have discussed the excess electron binding phenomenon and detachment process for giant fullerenes and nested fullerenes from a theoretical point of view using a simple one-electron/one-dimensional model. Although no experimental studies on this topic have been done, I think that this work may provide some insights into fullerene chemistry and physics. In conclusion, I think that the manuscript can be published once the following points would be addressed by the authors.

1.       The most important point of this study may be the validity of the one-electron/one-dimensional model used in the present study. Three parameters, namely, rinD, and U0, have been determined so as that the measured electron detachment energies can reproduce the measurements. Is this correct? Anyway, more detailed descriptions for the determination of these parameters should be given. No previous theoretical calculations (quantum chemistry calculations, first-principles calculations, or density-functional calculations) are available for the C60- anion, for example? From such results, the validity of the simple model used in this study could be discussed at a more quantitative level. If exist, the excess electron distribution functions can be available and the interaction between the excess electron and the valence electrons can be understood. If not, the authors can easily perform such calculations using modern quantum chemical calculation codes.

2.       On page 4, the authors state that “the charge density of the attached 1s-electron concentrates largely within the wall of a C60 (5.6 < r < 7.7 ) ….” I do not think that this is correct since one can see relatively large distributions of the excess electron in the tunneling regions outside of the potential well. I suggest that the authors can additionally show the potential energy curves of the one-dimensional model (+centrifugal potentials) to Figure 1. The authors can give a more quantitative description on the properties of the wavefunctions using the potential energy curves.

Reviewer 2 Report

Photodetachment cross sections of giant and nested fullerene anions are investigated theoretically using a simple potential model. While the photodetachment cross sections of fullerenes and nested fullerenes have been studied in several theoretical works, the knowledge of their anions is still limited. In addition, the present work using a simple model demonstrates the counter-intuitive results of the cross section behavior. These features are worth to be investigated in more detail in future works. Thus, the manuscript can be suitable for publication. I would like the authors to consider the following points before publication:

1) At eq. 3, please clarify the boundary conditions of the continuum state wavefunction, or strategy of calculation of photodetachment cross sections (e.g. utilizes the sum rule of the oscillator strength).

2) This might be my computer problem, but figure 1 is not visible in the manuscript. Please check the tex compilation.  

3) On the intriguing feature of P_3p, please consider having a short discussion on the centrifugal potential as the authors had for the similarity of P_2p and P_1s.

4) On the behavior of the photodetachment cross sections shown in figure 3. Since the final state is a continuum state of neutral (nested or bare) fullerene and an electron, the cross section should start at zero at the threshold energy due to the Wigner's law. It seems that the blue and red curves start at a finite value at the threshold energy. Please explain.
